# Diagnostic Accuracy of Multiplex Polymerase Chain Reaction in Early Onset Neonatal Sepsis

**DOI:** 10.3390/children10111809

**Published:** 2023-11-14

**Authors:** Anja Stein, Daniel Soukup, Peter-Michael Rath, Ursula Felderhoff-Müser

**Affiliations:** 1Department of Pediatrics I, Neonatology and Experimental Perinatal Neurosciences, Center for Translational and Behavioral Neuroscience, CTNBS, University Hospital Essen, Faculty of Medicine, University Duisburg-Essen, Hufelandstraße 55, 45147 Essen, Germany; daniel.soukup@chu-bordeaux.fr (D.S.); ursula.felderhoff-mueser@uk-essen.de (U.F.-M.); 2Service de Réanimation Pédiatrique, Centre Hospitalier Universitaire de Bordeaux, Place Amélie Raba Léon, 33000 Bordeaux, France; 3Institute for Medical Microbiology, University Hospital Essen, Hufelandstraße 55, 45147 Essen, Germany; peter-michael.rath@uk-essen.de

**Keywords:** neonatal sepsis, early onset sepsis, multiplex PCR, infection, neonate, diagnosis, pathogen detection

## Abstract

Early onset neonatal sepsis is a significant contributor to neonatal morbidity and mortality. Although blood cultures remain the diagnostic gold standard, they detect pathogens in only a minority of suspected cases. This study compared the accuracy of blood cultures with a rapid multiplex PCR test. Newborns at risk of neonatal sepsis were prospectively screened as recommended per national guidelines. Evaluations included laboratory parameters (CrP, IL6, differential blood count), blood culture, and a molecular multiplex PCR test (ROCHE LightCycler SeptiFast^®^) identifying 20 common microbial agents. Blood samples were taken simultaneously from umbilical cord or venous sources on the first day of life. Of 229 infants included, 69% were born preterm. Blood culture and multiplex PCR sensitivity were 7.4% and 14.8%, respectively. Specificity, negative and positive predictive values between methods showed no significant variance, although multiplex PCR had more false positives due to contamination. The limited sensitivity of blood cultures for early onset neonatal sepsis is concerning. Despite quicker results, multiplex PCR does not enhance diagnostic accuracy or antibiotic therapy guidance, thus it cannot be recommended for this indication.

## 1. Introduction

Neonatal sepsis remains a significant challenge in neonatology, escalating both morbidity and mortality [1]. Annually, an alarming three million children succumb to this condition [2]. Defined as manifesting within the first 72 h of life, early onset neonatal sepsis (EONS) primarily stems from infections ascending pre- or peripartum. This pathway elucidates why *Streptococcus agalactiae* (group B streptococci, GBS) and *Escherichia coli* are principal pathogens, attributing to nearly 70% of EONS cases. Documented incidence of culture-proven EONS ranges between 0.77 and 1 per 1000 deliveries, with mortality rates reaching 20% [1,3]. Notably, between 10 and 30% of pregnant women are colonized with GBS. Without maternal intrapartum antibiotic prophylaxis, neonates from these mothers display a colonization rate of 68%, and the risk of EONS attributed to GBS stands at 1.1% [4,5,6].

One of the significant hurdles in neonatal sepsis diagnosis is its non-distinct and occasionally subtle symptoms, particularly in preterm infants. Reliance on individual laboratory tests remains precarious due to their variable sensitivity and specificity, compelling a multi-parameter approach [7,8,9]. Blood cultures, although a diagnostic gold standard, often yield inconclusive results, particularly in neonates. The inoculation of at least 1 mL of venous blood for a blood culture would increase the sensitivity of pathogen detection but is rarely achieved in premature infants. Use of umbilical cord blood may be useful to ensure adequate blood volumes and can avoid skin punctures otherwise not indicated in the newborn but may be prone to contamination [10,11,12]. Moreover, the protracted 36–48 h waiting period for results, coupled with its diminished sensitivity and specificity in EONS, renders it less ideal [7]. Previous maternal antibiotic treatment and GBS-targeted intrapartum antibiotic prophylaxis further compromise bacterial detection [13]. EONS attributable to anaerobes, such as *Bacteroides*, is uncommon, yet can be associated with significant mortality. Presently, there are no commercially available anaerobic blood culture bottles validated for blood volumes under 3 mL. Consequently, according to the recommendation of the German national guideline [14], neonatal blood cultures are typically processed in aerobic bottles. The necessity and the role of anaerobic cultures in this context remain a subject of debate [15].

Molecular testing, especially polymerase chain reaction (PCR) and other nucleic acid amplification techniques such as metagenomic next-generation sequencing (NGS) technologies, offer potential advantages in diagnosing and managing neonatal sepsis [16]. Conventional cultures may require several days for definitive results, while molecular assays can yield outcomes within hours. Such swift diagnostics can facilitate the timely initiation of targeted antibiotic therapy, potentially enhancing patient outcomes [17]. Additionally, molecular techniques can discern small quantities of bacterial or fungal DNA or RNA, thereby potentially detecting infections that conventional cultures might overlook. However, this precision may also introduce the risk of false positives from potential contamination. As a result, it has been concluded that multiplex PCR testing provides no additional benefit in nosocomial and late-onset infections of preterm infants [18,19]. If antibiotics have been administered to mother or infant, traditional cultures may fail to cultivate the causative organism. However, molecular assays can identify the pathogen’s DNA or RNA even if the organism is rendered non-viable by antibiotics. Furthermore, some molecular tests can simultaneously identify a vast array of pathogens, enabling the detection of polymicrobial infections. As the specificity of the chosen test is contingent upon the pathogens incorporated in its panel, any organism not encompassed in the assay’s design may remain undetected. A noteworthy advantage of molecular tests is their reduced smaller blood volume requirement compared to traditional cultures [20]. The majority of such tests, however, cannot ascertain antibiotic susceptibility, an essential insight to guide the antimicrobial treatment that traditional cultures offer.

This research aimed to assess whether direct molecular testing, specifically the ROCHE LightCycler SeptiFast^®^—a multiplex PCR test that was commercially available at the initiation of the study—surpasses blood culture in accuracy and pathogen detection rate in EONS. This test targets sequences between bacterial 16S-23S ribosomal RNA and fungal 18S-5.6S ribosomal RNA [18]. Its scope, although limited to 20 select microorganisms (Table 1), covers pathogens implicated in 90% of adult and pediatric bloodstream infections [13]. Anaerobic bacteria are not represented in the LightCycler SeptiFast^®^ assay, yet it covers the majority EONS pathogens such as *Escherichia coli* and *Streptococcus agalactiae*. Notably, coagulase negative staphylococci, which are frequently detected as contaminant in blood cultures and PCR tests, are not a typical cause of EONS.

Given the potential advantages and limitations, molecular testing may be a valuable tool in the diagnostic algorithm for EONS, particularly when used in conjunction with clinical judgement and routine laboratory parameters.

## 2. Materials and Methods

Patient data were prospectively collected between March 2017 and September 2018. All neonates admitted to the neonatal intensive care unit at the Division of Neonatology of the Department of Pediatrics I at the University Hospital of Essen underwent a routine sepsis workup. Furthermore, healthy newborns screened for infection due to perinatal risk factors for neonatal sepsis, in accordance with the national S2k guideline for prophylaxis of early onset neonatal sepsis by GBS [14], were also eligible. The study received approval from the ethics committee of the University of Duisburg-Essen (16-7306-BO), and written informed consent was obtained from the legal guardians of the infants.

### 2.1. Laboratory Workup

Routine sepsis workup involved collecting a blood sample for culture, hematology, and clinical chemistry. Samples were collected within the first 72 h of life. For healthy newborns, umbilical cord blood was used to minimize the need for additional skin punctures for sampling. When neonates required peripheral blood sampling or the insertion of a peripheral venous catheter based on clinical indications, blood samples were primarily obtained from these venous sources. Blood collection followed disinfection using an alcohol-based antiseptic. Aerobic pediatric blood culture bottles (BD BACTEC Peds Plus /F^®^ blood culture bottles; Benex Limited, Dun Laoghaire, Ireland) were inoculated with a minimum of 0.5 mL of blood and then incubated for at least 7 days. Positive blood cultures were Gram stained, and detected microorganisms were further identified to the species level following standard microbiological methods. Additionally, 200 μL EDTA blood was employed for a compete blood count, and 300 μL of serum was used to determine CRP and Interleukin 6 levels. In addition, 100–200 μL EDTA blood was sampled for the multiplex-PCR (SeptiFast^®^) in DNA-free Sarstedt Microvette^®^ tubes (Nümbrecht, Germany). PCR analysis was commenced at the Institute of Medical Microbiology, University of Duisburg-Essen, using the LightCycler^®^ SeptiFast MGRADE system (Roche Diagnostics, Penzberg, Germany) with a modified DNA extraction protocol for small blood volumes [18,19]. The minimum dataset required was: blood culture, multiplex PCR, and CRP/IL-6. Patients with incomplete datasets were excluded from the study. Multiplex PCR and blood culture were considered positive if they detected at least one pathogen. PCR results were made available to clinicians within 6 to 12 h to guide therapeutic approaches in these patients.

### 2.2. Definition of EONS

To compare the results of blood culture and multiplex PCR, definition and confirmation of true EONS in our patients was mandatory. Diagnosis of EONS can be confirmed when a blood culture yields a typical pathogen in the context of clinical symptoms and laboratory indicators of infection. However, due to factors such as maternal antibiotic use, the timing and volume of blood sample collection, and the intermittent nature of bacteremia, a low sensitivity of the blood cultures and a correspondingly low rate of blood culture-proven EONS were expected. Consequently, EONS cannot be ruled out in the absence of a positive blood culture. While clinical late-onset or nosocomial sepsis have clear definitions based on both clinical and laboratory criteria (as outlined in the Surveillance Protocol NEOKISS by the Robert-Koch Institute (RKI), available at www.nrz-hygiene.de/KISS-Modul/KISS/NEO, accessed on 26 June 2023), a universally accepted definition for clinical EONS remains elusive [21]. Diagnosis of clinical EONS relies on a combination of clinical observations, laboratory results, and the medical histories of both the mother and the infant. We therefore adapted the risk stratification published by Stocker et al. [22] to define categories for “sepsis likely”, “sepsis possible”, or “sepsis unlikely”, providing clear criteria for assigning infants to each EONS risk category. In this system, one point is allocated for the fulfillment of any criteria within the three categories: anamnestic risk factors, clinical signs, and laboratory findings. This results in a potential maximum “clinical EONS score” of three points (as detailed in Table 2). The necessary anamnestic information of patients and mothers, clinical symptoms of infection, and laboratory parameters were extracted from medical records.

The patients were stratified into four distinct groups based on their blood culture results and clinical EONS score points (Table 3).

Our study did not include a control group in the traditional sense—that is, a group devoid of any clinical or laboratory signs or risk factors for infection (or zero EONS score points in Table 1). Since such infants would not typically be screened for infection according to our national guidelines, we did not have blood samples from such a true control group available for analysis.

Results from blood cultures and PCR testing were verified by the attending physician; *coagulase-negative staphylococci*, *Corynebacterium* species, *Bacillus* species, *Propionibacterium acnes*, *micrococci*, and *Neisseria* species other than *N. gonorrhoeae* and fungi were defined as possible contaminants. In groups two through four, blood culture results were either negative or deemed as false positives. Patients in groups one and two, categorized as “culture proven sepsis” and “sepsis likely”, respectively, were defined as having true EONS. The classifications were determined independent of multiplex PCR results. For groups one and two, positive PCR outcomes were viewed as potential true positives. Conversely, within groups three and four (“sepsis possible” or “sepsis unlikely”), all positive PCR outcomes were treated as false positives. 

### 2.3. Statistics

Data analysis and graphical displays were conducted using SPSS (version 28 for Mac) and Excel (version 16.53 for Mac). Continuous variables are presented as mean ± standard deviation (SD). Sensitivity, specificity, and positive and negative predictive values were derived using a confusion matrix.

## 3. Results

Between March 2017 and September 2018, a total of 2728 neonates were born at the Clinic for Obstetrics and Gynecology of the University Hospital in Essen. Of these, 900 were admitted to the neonatal intensive care unit (NICU) and neonatal intermediate care units on their first day of life. Additionally, 20 outborn neonates were admitted to these units during the same period. After obtaining parental consent, 229 neonates with a complete laboratory set were included in the study (birth-weight range: 400 g to 4750 g; gestational age: range: 23 + 4/7 to 41 + 4/7 weeks). The study cohort comprised 40 very low birth-weight infants (<1500 g) and 159 preterm infants, of which 43 were very preterm infants born before 32 weeks of gestation. The distribution of patients by gestational age within the EONS groups is detailed in Table 4.

### 3.1. Results from Blood Cultures

Of all blood cultures taken, six tested positive. However, only two of these were deemed true positives. *Escherichia coli* was identified in one very preterm infant, and *Streptococcus agalactiae* in one term infant. Both these infants exhibited clinical and laboratory signs of infection. This results in an EONS rate of 0.9% in this cohort.

Four cultures were judged as contaminants because the corresponding infants did not manifest any clinical or laboratory signs of infection, did not receive antibiotic treatment, and were screened for infection solely due to anamnestic risk factors (EONS risk group 4). Two of the assumed contaminated cultures revealed growth of *Staphylococcus epidermidis*, an organism not typically associated with EONS. One culture identified multiple pathogens (*Streptococcus mitis*, *Escherichia coli*, and *Enterococcus faecalis*), further suggesting contamination. One culture yielded *Klebsiella pneumoniae*, which could have been a potential true pathogen for EONS.

Maternal antibiotic treatment prior to delivery may influence the growth of pathogens in blood cultures obtained from infants postnatally. Prenatal maternal antibiotic treatment was documented in 48 out of 229 infants (20.1%). Among the six positive blood cultures, four were obtained from infants without prenatal maternal antibiotic therapy. Among these were the two true positive blood cultures. Aside from premature birth of one infant the mothers of these two infants did not exhibit any risk factors for EONS such as prolonged rupture of membranes, GBS colonization or signs of chorioamnionitis.

### 3.2. Results from Multiplex PCR Tests

Multiplex PCR testing failed to detect pathogens in the two instances of blood culture-proven sepsis. Both infants were born to mothers without premature or prolonged rupture of membranes, without clinical signs of amnion infection, and without antibiotic treatment prior to giving birth.

Multiplex PCR testing identified pathogens in 19 cases from patients in EONS groups two to four. Of all multiplex PCR tests, only two were deemed potentially true positives, and both fell under the EONS category “sepsis likely”.

The infants within the EONS group “sepsis likely” (*n* = 25) displayed clinical and laboratory signs of sepsis in conjunction with anamnestic risk factors. The majority of patients in EONS group two had respiratory symptoms suggestive of infection (22 of 25 patients). The remaining three patients had: temperature instability, apathy, and mottled skin color. Within the first 24 h of life 15 of the patients showed laboratory signs of infection (leukopenia < 5/nL in *n* = 4, CrP > 5 mg/L in *n* = 5, IL-6 > 100 pg/mL in *n* = 12, I/T ratio > 0.2 in *n* = 1). The remaining 10 patients showed elevated CrP or IL-6 levels after 24 to 72 h of life (CrP > 5 mg/L *n* = 9, IL-6 > 100 pg/mL *n* = 2). Elevated CrP levels within the first 24 h ranged from 7 to 44 mg/L, and 9 to 108 mg/L after 24 to 72 h. Elevated IL-6 levels ranged from 200 to 12,890 pg/mL. Choosing higher cut-off values, the maximum CrP level remained ≤ 20 mg/L in 14 patients and ≤10 mg/L in 8 patients from this group, among them were the two patients with the highest IL-6 levels (both > 12,000 pg/mL). Anamnestic risk factors within this subgroup were: prolonged rupture of membranes > 18 h (*n* = 5), clinical signs of chorioamnionitis (*n* = 7), and/or premature birth (*n* = 17). Maternal colonization status with *Streptococcus agalactiae* was unknown in *n* = 17 (negative in *n* = 7). Within the EONS group “sepsis likely”, none of the blood cultures were positive, but four multiplex PCR tests were. Potentially true pathogens of EONS identified were *Staphylococcus aureus* in a very preterm infant and *Enterobacter cloacae* in a term infant. The preterm infant initially received antibiotic treatment with ampicillin and gentamycin. This was later adapted to a 3rd generation cephalosporin due to the discovery of resistant *Klebsiella pneumoniae* in the maternal vaginal swabs and a rising CrP level reaching 108 mg/L. The antibiotic treatment regimen was not adapted to specifically target *Staphylococcus aureus* despite the multiplex PCR result. The term infant, presenting with an initial IL6 level of 4015 pg/mL and a CrP of 28 g/dL, was administered ampicillin for 8 days and gentamycin for 6 days. The treatment regimen remained unchanged even after obtaining the PCR result, notwithstanding the intrinsic resistance of *Enterobacter cloacae* to aminopenicillins owing to the production of constitutive AmpC β-lactamase. The other two tests detected *coagulase-negative staphylococci* and *Candida tropicalis*, both likely contaminants. The maximal CrP level in both infants remained below 20 mg/L, even though the IL-6 level in one of the infants was markedly elevated (12,890 pg/mL). In this EONS group, six mothers had displayed signs of amnion infection or prolonged rupture of membranes > 18 h and received antibiotic treatment prior to giving birth (6/25 mothers, 24%). Only one of the six infants born to these mothers had a positive PCR result (the preterm infant mentioned above, *Staphylococcus aureus*).

For the EONS group “sepsis possible”, multiplex PCR detected positive results in four patients, all of whom had anamnestic risk factors for infection. Two tests identified *Candida krusei*, presumed to be contaminants. Neither of these patients received antimycotic treatment. One showed potential clinical signs but lacked laboratory evidence of infection, while the other displayed no clinical symptoms but had an elevated interleukin 6 level at 321 pg/mL as the singular laboratory sign. The last two tests from this group were also considered contaminated: one identified *coagulase-negative staphylococci*, while the other detected both *coagulase-negative staphylococci* and *Staphylococcus aureus*. Notably, the latter very preterm infant had congenital cutaneous candidiasis and received appropriate systemic antimycotic treatment.

In the EONS group “sepsis unlikely”, eleven patients returned positive results on the multiplex PCR test. Detected pathogens included *Candida albicans* (*n* = 1) and *coagulase-negative staphylococci* (*n* = 6), neither of which are typical for EONS. Other identified pathogens, namely *Streptococcus species*, *Staphylococcus aureus*, *Klebsiella*, and *Enterobacter cloacae*, have potential associations with EONS. However, none of these patients exhibited clinical symptoms, and none received antibiotic treatment.

### 3.3. Antibiotic Treatment

Classification into the EONS groups corresponded with the administration and duration of antibiotic therapy. All patients in EONS groups one and two (*n* = 27) received antibiotics with a mean treatment duration of 7.2 ± 5.0 days. Notably, 12 out of 27 patients (44.4%) were very preterm infants, who are often administered antibiotics more liberally. Within the “sepsis likely” EONS group two, the four patients with positive multiplex PCR tests had a longer treatment duration compared to those without pathogen detection (7.3 ± 2.2 days vs. 6.2 ± 3.3 days, *p* = 0.002). However, the chosen antibiotic regimen was not targeting the identified pathogens.

In the “sepsis likely” category (EONS group 3), 91.9% of infants (34 out of 37) received antibiotics, with an average treatment duration of 4.8 ± 4.9 days. This group included 27% very preterm infants (10/37), all of whom were given antibiotic treatment. The duration of antibiotic therapy did not significantly differ between patients who had pathogen detection through PCR testing and those who did not (4.8 ± 1.7 days vs. 4.8 ± 5.2 days, respectively). All patients who tested positive in PCR were administered antibiotics.

For the EONS group with the lowest risk of infection (EONS group 4), only 7.3% of patients (12 out of 165) underwent antibiotic treatment, with an average duration of 5.1 ± 9.0 days. Notably, 11 out of these 12 infants were treated for 3 days or less, reinforcing the belief that they likely did not have EONS. None of the patients with either a positive PCR test result or a positive blood culture received antibiotic treatment. This cohort predominantly comprised late preterm infants (97 out of 165, or 58.8%), with a smaller fraction being very preterm infants (21 out of 165, or 12.7%).

### 3.4. Comparison of Reference and Index Test

Defining patients in group one (positive blood culture) and patients in group two as gold standard and having true EONS, the confusion matrix revealed no significant differences between the reference (blood culture) and the index test (multiplex PCR). As Table 5 illustrates, both tests exhibited very low sensitivity. Other metrics showed minimal variation between the two tests. Both achieved a specificity of over 90%, with the blood culture being marginally superior. The positive predictive value exhibited a broad confidence interval, rendering it an unreliable metric. The negative predictive values were identical for both tests. Although four multiplex PCR tests were positive in patients who were defined as having true EONS, leading to a sensitivity of 14.8%, two of these were clinically judged as contaminations and only two detected potential pathogens for EONS.

## 4. Discussion

The incidence of culture-proven EONS in our study was 0.9%, which is higher than in the general neonatal population where it is 0.1% of all live births [3,23]. This discrepancy can be attributed to a selection bias since our study included a high proportion of preterm infants as well as infants who required admission to a neonatal intensive care unit. In cohorts comprising very low birth-weight infants, the incidence is anticipated to be around 1.3–1.4% [23,24]. Nonetheless, this selection bias likely does not affect the study’s results. 

The known low sensitivity of blood culture in EONS is significantly influenced by factors such as blood volume, the number of samples, timing, and maternal antibiotic treatment [25,26]. Also, other maternal conditions and their treatments—such as preeclampsia, autoimmune diseases, or maternal cancer—could potentially influence the neonate’s inflammatory response and the likelihood of infection Due to the considerable variation and lack of homogeneity in these factors within our cohort, we did not conduct a detailed evaluation of maternal illnesses or drugs, since this precluded a robust statistical analysis. Within our study, only two infants were diagnosed with culture-proven sepsis, both identified with typical pathogens: *Escherichia coli* and *Streptococcus agalactiae*. A limitation we encountered was that we likely did not consistently achieve the recommended pediatric blood culture volume of 1 mL, especially in very low birthweight infants. This exact volume was not documented. Our internal guidelines permitted 0.5 to 1 mL for venous samples in premature infants. While this can diminish the sensitivity of blood cultures, it likely mirrors prevalent clinical practice.

Notably, the multiplex PCR in our study failed to detect the pathogens identified in the blood culture. Previous studies suggest that molecular testing can yield reliable results with much smaller blood quantities, even in cases of low or intermittent neonatal bacteremia or previous maternal antibiotic treatment [19,20]. This potential advantage could account for the higher rate of false positives due to PCR sample contamination. In the EONS group labeled “sepsis unlikely”, four blood cultures and 11 PCR tests were deemed to be contaminants. A significant proportion of these patients were primarily screened for infection based on anamnestic risk factors, displayed no symptoms, and likely had samples drawn from their umbilical cord blood instead of venous blood. This method was preferred when there was no clinical justification for venous puncture in the newborn. While umbilical cord samples might be more voluminous due to easier access, less stringent aseptic conditions during sampling could lead to a higher contamination rate. A notable limitation is that the source of the blood was not tracked during our study. Although Diericks et al. [12] argued that “umbilical cord blood culture has higher sensitivity and comparable specificity for the diagnosis of neonatal early-onset sepsis”, such findings might not be directly applicable to this low-risk EONS group. 

Sensitivity of the multiplex PCR test in adult studies ranges from 60 to 95% and specificity between 74 and 99% [13]. In children of all ages, including neonates, sensitivity ranges from 85 to 90.2% and specificity between 72.9 and 93.5 % [17,19]. This can be confirmed by the present study reporting a specificity of 94.6% for the multiplex PCR but with a higher specificity for the blood culture with 98.0%. In a study of late onset neonatal sepsis, the false positive rate of the multiplex PCR test was as high at 27.1 to 35% [18,19]. Although the rate of false positive tests of 5.4% in our study of EONS is lower than this, the multiplex PCR achieved a sensitivity of only 14.8% and low positive and negative predictive values (26.7% and 89.3%). This does not make it a reliable test to diagnose or exclude EONS.

One theoretical advantage of multiplex PCR testing is the availability of results within 6 to 24 h depending on the laboratory setting. Blood culture results can be safely interpreted after 36 to 48 h, but the clinicians’ trust in sterile cultures is low [7,8]. As there is a lack of other reliable laboratory indicators for EONS and overlap of symptoms with non-infectious neonatal conditions like respiratory distress or hypoxia ischemia, this has potentially led to an overexposure to antibiotics. It is recommended to stop empirical antibiotic treatment if blood cultures remain sterile after 36 to 48 h. Therefore, reliable PCR results could potentially reduce unnecessary antibiotic exposure. Among the patients in groups 1 and 2, 25 of 27 blood cultures (93%) returned were sterile, although all the patients were clinically likely to have EONS. Four patients had positive multiplex PCR findings; however, the pathogens detected in two cases were not typical causes of EONS, and none of the patients with positive blood cultures had corresponding PCR results. Ultimately, the PCR results did not influence initiation, choice, or duration of antibiotic treatment in any of the cases in our cohort, although they were available to the treating clinicians.

All neonates within EONS group “sepsis likely” received antibiotic therapy irrespective of negative blood culture results. The duration of antibiotic therapy correlated with the EONS risk group stratification. Of note is that 7.3% of patients in the group “sepsis unlikely” received antibiotics with a mean duration of 2.8 days. No antibiotic therapy was initiated because of a positive PCR, but the duration of the antibiotic therapy was longer than in the patients without any microbiological proof, even in group 3 (“sepsis unlikely”). Overuse of antibiotics may have negative effects, especially in preterm infants, in whom prolonged antibiotic therapy increases the mortality and risk for bronchopulmonary dysplasia, retinopathy, necrotizing enterocolitis, and damage to periventricular white matter [11]. Multiplex PCR testing is not suitable to increase diagnostic security, especially in a setting of low-risk infants that are only screened for infection (EONS group “sepsis unlikely”).

The multiplex PCR test which was used in our study (LightCycler^®^ SeptiFast MGRADE system, Roche Diagnostics, Penzberg, Germany) is no longer commercially available. Even as metagenomic NGS methods emerge that do not necessitate specific primer design, allowing for the detection of a broad spectrum of bacterial, fungal, and viral pathogens in one assay [27,28,29] and have the potential to detect antimicrobial resistance genes, we believe our study’s results would remain largely consistent. Even though the multiplex PCR we employed already encompassed the most common pathogens associated with EONS, the diagnostic yield was not enhanced for culture-negative patients diagnosed with EONS based on clinical, laboratory, and anamnestic indicators (EONS group two). However, given the non-specific nature of clinical symptoms and the relatively low specificity of inflammatory markers, particularly on the first day of life or in conditions such as asphyxia, even the combination of laboratory, clinical and anamnestic signs cannot prove EONS with all certainty. Definition of EONS is also highly dependent on the chosen cut-off values of inflammatory markers as well as other factors such as timing of blood withdrawal or delivery mode [21,30]. Consequently, it is possible, that not all infants in EONS group two had a bacterial blood stream infection detectable by blood culture, multiplex PCR, or even NGS methods. Clinicians must ensure, however, that infants with possible sepsis are not missed and are provided with timely and potentially life-saving antimicrobial therapy. As a result, all infants in EONS group two were treated with antibiotics.

The primary objective of any antibiotic stewardship program is threefold: to prevent the onset of unnecessary antibiotic therapy, to cease empirical antibiotic treatment once an infection can be confidently excluded, and to pinpoint and manage neonates with sepsis using precisely targeted antimicrobial therapy. Even with the promising advances in next-generation sequencing (NGS) and other molecular techniques, the search for the perfect diagnostic marker persists.

Our research underscores that, especially in the context of infection screening within a low-risk cohort, multiplex PCR testing is not optimal for guiding EONS diagnosis. Therefore, a swift marker boasting high sensitivity, specificity, and predictive value remains a sought-after aim.

## 5. Conclusions

Our research demonstrates that multiplex PCR testing did not enhance pathogen detection, diagnostic accuracy, or guide antibiotic therapy in early-onset neonatal sepsis (EONS) compared to standard blood culture. Multiplex PCR failed to detect pathogens found in two positive blood cultures, exhibiting a very low sensitivity (14.8%), similar to blood culture. Specificity and negative and positive predictive values showed no significant differences between PCR and blood culture. PCR yielded more false positives, likely due to sampling contamination, primarily involving pathogens not associated with EONS. Rapid PCR results did not influence antibiotic therapy decisions, with therapy duration correlating well with clinical EONS risk stratification. Given its low sensitivity and potential for false positives, multiplex PCR therefore cannot be recommended for EONS diagnosis or exclusion, particularly in low-risk screening settings. More reliable markers for neonatal sepsis are still warranted.

## Figures and Tables

**Table 1 children-10-01809-t001:** Pathogens included in the assay of the ROCHE LightCycler SeptiFast^®^ [19].

Gram-Negative Organisms	Gram-Positive Organisms	Fungi
*Escherichia coli*	*Staphylococcus aureus*	*Candida albicans*
*Klebsiella pneumoniae/oxytoca*	*Coagulase negative staphylococci* ^1^	*Candida tropicalis*
*Serratia marcescens*	*Streptococcus pneumoniae*	*Candida parapsilosis*
*Enterobacter cloacae/aerogenes*	*Streptococcus* ssp. ^2^	*Candida glabrata*
*Proteus mirabilis*	*Enterococcus faecium/faecalis*	*Candida krusei*
*Pseudomonas aeruginosa*		*Aspergillus fumigatus*
*Acinetobacter baumannii*		
*Stenotrophomonas maltophilia*		

^1^ *S. epidermidis*, *S. haemolyticus*, *S. hominis*, *S. pasteuri*, *S. warneri*, *S. cohnii*, *S. lugdunensis*, *S. capitis*, *S. caprae*, *S. saprophyticus*, and *S. xylosus*. ^2^ *S. agalactiae*, *S. pyogenes*, *S. anginosus*, *S. bovis*, *S. constellatus*, *S. cristatus*, *S. gordonii*, *S. intermedius*, *S. milleri group*, *S. mitis*, *S. mutans*, *S. oralis*, *S. parasanguinis*, *S. salivarius*, *S. sanguinis*, *S. thermophilus*, *S. vestibularis*, *S. viridans group*.

**Table 2 children-10-01809-t002:** Clinical EONS Score adapted from [22].

Category	Criteria
anamnestic risk factors for EONS	chorioamnionitis (maternal fever > 38.5 °C, maternal or fetal tachycardia, uterine tenderness, malodorous vaginal discharge, and maternal leukocytosis ≥ 15 /nL) prolonged rupture of membranes > 18 h prematurity (birth below 37 weeks of gestation) maternal colonization by GBS * GBS * bacteriuria during gravidity GBS * sepsis in a neonate from a previously pregnancy
clinical criteria	fever > 38 °C, hypothermia < 36.5 °C or temperature instability tachycardia > 200/min or frequent bradycardia < 80/min arterial hypotension, poor perfusion (recapillarization time > 2 s) tachypnea > 60/min, dyspnea, frequent desaturations, or apnea vomiting, feeding intolerance irritability or seizure apathy, lethargy, mottled skin color, general instability
laboratory criteria	C-reactive protein (CrP) > 5 mg/L
	Interleukin-6 > 100 pg/ml
	Ratio of immature to mature neutrophils (I/T ratio) > 0.2 White blood cells < 5 /nL

* GBS—group B streptococci, Streptococcus agalactiae.

**Table 3 children-10-01809-t003:** Study groups based on blood culture result and EONS Score.

EONS Group	Blood Culture	EONS Score Points
1: culture proven sepsis 2: sepsis likely 3: sepsis possible 4: sepsis unlikely	true positive false positive or negative false positive or negative false positive or negative	1–3 3 2 0–1

**Table 4 children-10-01809-t004:** Patient distribution and demographic data in the EONS Study groups.

Study Group	Total	GA > 36 + 6/7 Weeks (Term)	GA 32 + 0/7 to 36 + 6/7 Weeks (Late Preterm)	GA < 32 + 0/7 Weeks (Very Preterm)
Total number of patients	229	70 (30.6%)	116 (50.7%)	43 (18.8%)
Mean GA ± SD (weeks) Mean birth weight ± SD (g)	35.3 ± 3.9 2370 g ± 879	38.7 ± 1.3 3230 ± 546	34.9 ± 1.2 2270 ± 495	28.4 ± 2.4 1055 ± 446
1: culture proven sepsis 2: sepsis likely 3: sepsis possible 4: sepsis unlikely	2 (0.9%) 25 (10.9%) 37 (16.2%) 165 (72.1%)	1 7 15 47	0 7 12 97	1 11 10 21

GA—gestational age, SD—standard deviation.

**Table 5 children-10-01809-t005:** Comparison of sensitivity, specificity, negative and positive predictive values between reference test (blood culture), and index test (multiplex PCR).

	Blood Culture		Multiplex PCR	
	Estimated Value (%)	95% Confidence Interval (%)	Estimated Value (%)	95% Confidence Interval (%)
Sensitivity	7.4	1.3–25.8	14.8	5.9–23.7
Specificity	98.0	94.7–99.4	94.6	91.8–97.3
Positive predictive value	33.3	6.0–75.9	26.7	7.9–45.4
Negative predictive value	88.8	83.7–92.5	89.3	84.8–93.7

## Data Availability

The dataset used and/or analyzed for the study is available from the corresponding author upon reasonable request.

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
