# Peer review of "Diagnostic Accuracy of Multiplex Polymerase Chain Reaction in Early Onset Neonatal Sepsis"

_children, 2023, doi:10.3390/children10111809_

Round 1

Reviewer 1 Report

Comments and Suggestions for Authors

The topic of the present paper Diagnostic accuracy of multiplex PCR in early onset neonatal sepsis is very interesting for readers, this study comparing the accuracy of blood cultures with a rapid multiplex PCR test, knowing that neonatal sepsis remains a significant challenge in neonatology, escalating both morbidity and mortality.

The authors show that molecular testing may be a valuable tool in the diagnostic algorithm for early onset neonatal sepsis, particularly when used in conjunction with clinical judgement and routine laboratory parameters.

The data presented underscore that, especially in the context of infection screening within a low-risk cohort, multiplex PCR testing is not optimal for guiding early onset neonatal sepsis diagnosis.

Finally, I conclude that:

-         1. the introduction provides sufficient background and includes relevant references;

-         2. the work design is well described;

-         3. the reference list is relatively recently;

-         4. the manuscript is well written, and the text is easy to read.

Author Response

Response to reviewer 1

1. Summary

Thank you for taking the time to review our manuscript titled "Diagnostic accuracy of multiplex PCR in early onset neonatal sepsis" and for providing your valuable feedback.

2. Questions for General Evaluation

Reviewer’s Evaluation

Does the introduction provide sufficient background and include all relevant references?

Yes

Are all the cited references relevant to the research?

Yes

Is the research design appropriate?

Yes

Are the methods adequately described?

Yes

Are the results clearly presented?

Yes

Are the conclusions supported by the results?

Yes

3. Point-by-point response to Comments and Suggestions for Authors

The topic of the present paper Diagnostic accuracy of multiplex PCR in early onset neonatal sepsis is very interesting for readers, this study comparing the accuracy of blood cultures with a rapid multiplex PCR test, knowing that neonatal sepsis remains a significant challenge in neonatology, escalating both morbidity and mortality.

The authors show that molecular testing may be a valuable tool in the diagnostic algorithm for early onset neonatal sepsis, particularly when used in conjunction with clinical judgement and routine laboratory parameters.

The data presented underscore that, especially in the context of infection screening within a low-risk cohort, multiplex PCR testing is not optimal for guiding early onset neonatal sepsis diagnosis.

Finally, I conclude that:

-         1. the introduction provides sufficient background and includes relevant references;

-         2. the work design is well described;

-         3. the reference list is relatively recently;

-         4. the manuscript is well written, and the text is easy to read.

We are pleased to hear that you find the topic interesting and pertinent to the readers. We hope therefore our manuscript will be deemed suitable for publication.

Reviewer 2 Report

Comments and Suggestions for Authors

Dear Authors,

The article presents an interesting research that approaches a less common point of view, to demonstrate the reduced efficiency of a PCR method for the evaluation and diagnosis of early sepsis in newborns. It is a study that supports the sensitivity and specificity of the PCR test compared to these factors in adults. But, the conclusions section is missing from the article, they can be found partially in the abstract and in the discussion section, but for a clearer picture it is necessary for the authors to differentiate the conclusions section.

In the list of pathogenic agents in table 1, no anaerobes are included, therefore it would be useful for the authors to specify why they are not included, even if their role in sepsis is quite unclear, however, mortality especially through Bacteroides bacteremias, it is quite important.

The article is well structured, with the description of the sections in a coherent and explicit way that makes it accessible and useful.

Author Response

Response to reviewer 2

1. Summary

Thank you very much for taking the time to review this manuscript. Please find the detailed responses below and the corresponding revisions highlighted in the re-submitted file.  

2. Questions for General Evaluation

Reviewer’s Evaluation

Response and Revisions

Does the introduction provide sufficient background and include all relevant references?

Yes 

Are all the cited references relevant to the research?

Can be improved

Is the research design appropriate?

Can be improved

Are the methods adequately described?

Yes

Are the results clearly presented?

Can be improved

Are the conclusions supported by the results?

Must be improved

A conclusion section was added to the re-submitted paper.

  3. Point-by-point response to Comments and Suggestions for Authors

Comment 1:

The article presents an interesting research that approaches a less common point of view, to demonstrate the reduced efficiency of a PCR method for the evaluation and diagnosis of early sepsis in newborns. It is a study that supports the sensitivity and specificity of the PCR test compared to these factors in adults. But, the conclusions section is missing from the article, they can be found partially in the abstract and in the discussion section, but for a clearer picture it is necessary for the authors to differentiate the conclusions section.

Response to comment 1: 

We acknowledge this oversight on our part. While we did incorporate key findings in the abstract and discussion, we understand the importance of presenting a clear, concise conclusions section to encapsulate the main outcomes and implications of our research for our readers.

To address this, we drafted a comprehensive conclusions section that distinctly summarizes our study's main findings and their implications for clinical practice. We agree this will provide clarity and make our manuscript more reader-friendly and impactful.

Comment 2:

In the list of pathogenic agents in table 1, no anaerobes are included, therefore it would be useful for the authors to specify why they are not included, even if their role in sepsis is quite unclear, however, mortality especially through Bacteroides bacteremias, it is quite important.

Response to comment 2:

Thank you for pointing this out. The multiplex PCR test, which was used in our study (LightCycler® SeptiFast MGRADE system, Roche Diagnostics), did not encompass anaerobes. This was the only commercially available multiplex PCR kit at the initiation of the study. Modern NGS methods may include anaerobes, but EONS due to anaerobes is rare and only few publications are found in the literature. In the retrospective study of Chatue Kamga (Neonatal early onset sepsis due to anaerobies: myth or realities: a review of medical record in one neonatal centre. J Clin Neonatol. 2013 Apr;2(2):110-1. doi: 10.4103/2249-4847.116416) no proven EONS infection was observed. Only two neonates with probable EONS (clinical and laboratory signs) but negative blood culture had a positive culture of anaerobes in their gastric aspirate. Thus it is unlikely that the inclusion of anaerobes in the PCR test would have altered the general results of our study. However, in light of your feedback, we recognize the importance of addressing this gap. To address this issue, we added a brief section in the manuscript discussing the role of anaerobes in sepsis, their diagnostic challenges, and the significance of associated bacteremias.

Thank you for deeming our article structured, accessible and useful.

Reviewer 3 Report

Comments and Suggestions for Authors

Dear Editor and Authors,

Thank you for offering the opportunity to evaluate the article titled "Diagnostic Accuracy of Multiplex PCR in Early Onset Neonatal Sepsis".

In this study, compared the use of blood cultures and the use of Multiplex PCR test for accuracy of diagnosis of sepsis in newborns with risk of early sepsis.

Correction of the following points will be useful for the understanding of the article:

Definitions related to sepsis should be made in accordance with international literature: Proven sepsis, clinical sepsis or sepsis likely, control group etc.

Maternal antibiotic use should be specified: The babies of the mothers who are given antibiotics for PROM or IAP have been taken to the study? Maternal antibiotic use reduces the likelihood of breeding in blood culture in infants with early neonatal sepsis.

Information should be given about the results of the laboratory examinations used in the diagnosis of sepsis: CBC, IT ratio, CRP, IL-6 etc.

The demographic and neonatal characteristics of term and preterm newborns in the study should be given separately.

Maternal diseases, drugs used should be given.

It should be stated whether the multiplex PCR method has a benefit on the decisions of cutting antibiotics to infants.

The diagnosis of sepsis in newborns is based on clinical symptoms and findings. In a newborn diagnosed with sepsis according to clinical symptoms and findings, the lack of bacterial growth in blood culture and/or the normal laboratory sepsis markers does not exclude the sepsis. In this study, how do the authors interpret the lack of bacterial growth in blood culture and/or the normal multiplex PCR test in infants where sepsis cannot be excluded according to clinical findings or high risk factors? The results of the study must be interpreted according to clinical symptoms and sepsis risk factors. For this reason, this study will provide more accurate interpretation of the working method used to include newborns with late onset sepsis.

Best regards

Author Response

Response to Reviewer 3 Comments

1. Summary

Thank you very much for taking the time to review this manuscript. Please find the detailed responses to your comments and suggestions below. The corresponding revisions are highlighted in the re-submitted files.

2. Questions for General Evaluation

Reviewer’s Evaluation

Response and Revisions

Does the introduction provide sufficient background and include all relevant references?

Yes

Are all the cited references relevant to the research?

Yes

Is the research design appropriate?

Can be improved

See response to comment 1 and 4

Are the methods adequately described?

Can be improved

See response to comment 1 and 4

Are the results clearly presented?

Can be improved

See response to comment 2 to 4

Are the conclusions supported by the results?

Can be improved

See response to comment 5-7

3. Point-by-point response to Comments and Suggestions for Authors

Comment 1: Definitions related to sepsis should be made in accordance with international literature: Proven sepsis, clinical sepsis or sepsis likely, control group etc.

Response to comment 1: Thank you for your comment. Early-onset neonatal sepsis (EONS) is defined as a bloodstream infection that occurs within the first 72 hours of life. Diagnosis of EONS can be confirmed when a blood culture yields a typical pathogen in the context of clinical symptoms and laboratory indicators of infection. However, due to factors such as maternal antibiotic use, the timing and volume of blood sample collection, and the intermittent nature of bacteremia a low sensitivity of blood cultures has been described in the literature. Additionally, the non-specific nature of clinical symptoms and the relatively low specificity of inflammatory markers, especially on the first day of life, further complicate the diagnosis of EONS. Consequently, EONS cannot be ruled out in the absence of a positive blood culture. Diagnosis relies on a combination of clinical observations, laboratory results, and the medical histories of both the mother and the infant. We have adopted the risk stratification method published by Stocker et al. to define categories for 'sepsis likely,' 'sepsis possible,' or 'sepsis unlikely,' providing clear criteria for assigning infants to each EONS risk category. Our study did not include a control group in the traditional sense — that is, a group devoid of any clinical or laboratory signs or risk factors for infection. Since such infants would not typically be screened for infection according to our national guidelines, we did not have blood samples from a true control group available for analysis. We clarified this in the methods section: 2.2 Definition of EONS (pages 4).

Comment 2: Maternal antibiotic use should be specified: The babies of the mothers who are given antibiotics for PROM or IAP have been taken to the study? Maternal antibiotic use reduces the likelihood of breeding in blood culture in infants with early neonatal sepsis.

Response to comment 2: We agree with this comment. Information on maternal antibiotic treatment prior to birth has been added to section 3.1 in the results (page 6).

Comment 3: Information should be given about the results of the laboratory examinations used in the diagnosis of sepsis: CBC, IT ratio, CRP, IL-6 etc.

Response to comment 3: Thank you for this suggestion. For enhanced clarity the and better characterization of EONS group 2 the results from the laboratory examinations have been included on page 6 (Section 3.2)

Comment 4: The demographic and neonatal characteristics of term and preterm newborns in the study should be given separately.

Response to comment 4: Thank you for this suggestion. For enhanced clarity the demographic data for the term and preterm subgroups have been included in Table 4 on page 5.

Comment 5: Maternal diseases, drugs used should be given.

Response to comment 5: Thank you for mentioning this. Although maternal conditions and their treatments—for instance, preeclampsia, autoimmune diseases, or maternal cancer—could potentially influence the neonate's inflammatory response and the probability of infection, we did not conduct a detailed evaluation of maternal illnesses or drugs. This is due to the considerable variation and lack of homogeneity in these factors within our cohort, which precluded a robust statistical analysis. We added this as a limitation of our study (page 8)

Comment 6: It should be stated whether the multiplex PCR method has a benefit on the decisions of cutting antibiotics to infants.

Response to comment 6: Thank you for pointing this out. The ultimate goal of any antibiotic stewardship program is to limit unnecessary antibiotic exposure. Rapid detection (or non-detection) of pathogens could support clinicians in choosing appropriate antibiotics or in discontinuing them as early as safely possible. Unfortunately, in our cohort none of the PCR results influenced initiation, duration, or choice of antibiotic treatment. We already elaborated on this in the discussion, but emphasized this point following your suggestion (page 9)

Comment 7: The diagnosis of sepsis in newborns is based on clinical symptoms and findings. In a newborn diagnosed with sepsis according to clinical symptoms and findings, the lack of bacterial growth in blood culture and/or the normal laboratory sepsis markers does not exclude the sepsis. In this study, how do the authors interpret the lack of bacterial growth in blood culture and/or the normal multiplex PCR test in infants where sepsis cannot be excluded according to clinical findings or high risk factors? The results of the study must be interpreted according to clinical symptoms and sepsis risk factors. For this reason, this study will provide more accurate interpretation of the working method used to include newborns with late onset sepsis.

Response to comment 7: Thank you for mentioning this very important point. How can we prove bacterial blood stream infections in neonates? EONS and late-onset neonatal sepsis cannot safely be excluded in the absence of a positive blood culture when clinical or laboratory indicators suggest a potential infection. Clinicians must ensure that infants with possible sepsis are not missed and are provided with timely and potentially life-saving antimicrobial therapy. Diagnosis of EONS and treatment decisions rely on a combination of clinical symptoms, laboratory markers of inflammation, and risk factors. However, given the non-specific nature of clinical symptoms and the relatively low specificity of inflammatory markers, particularly on the first day of life or in conditions such as asphyxia, even the combination of laboratory, clinical and anamnestic risk factors cannot diagnose EONS with all certainty. Consequently, it is possible, that not all infants in group 2 had a bacterial blood stream infection detectable by blood culture, multiplex PCR, or even NGS methods. We added this interpretation of our results (page 9).

To better characterize EONS group 2 we added information, which clinical, laboratory and anamnestic risk factors from the stratification method published by Stocker et al. lead to the classification “EONS likely” (page 6).

Incorporating your suggestions has undoubtedly strengthened our paper, and we believe that the revised manuscript now offers a more comprehensive overview of our results and a more detailed and nuanced discussion. We are confident that these changes will improve the overall quality of our research.

Round 2

Reviewer 3 Report

Comments and Suggestions for Authors

Dear Editor,

Dear Authors,

I believe that the publication of the revised article of this study, which shows that the multiplex PCR method, which allows the identification of the causative agent in neonatal sepsis in a short time, does not provide a diagnostic contribution, will contribute to the literature. Multiplex PCR method may contribute to the exclusion of sepsis and shortening the duration of antibiotics in babies who are started on antibiotics.

Best regards